# Health Education Module Based on Information–Motivation–Behavioural Skills (IMB) for Reducing Depression, Anxiety, and Stress among Adolescents in Boarding Schools: A Clustered Randomised Controlled Trial

**DOI:** 10.3390/ijerph192215362

**Published:** 2022-11-21

**Authors:** Rahmat Dapari, Mohd Safrin Mohamad Bashaabidin, Mohd Rohaizat Hassan, Nazri Che Dom, Syed Sharizman Syed Abdul Rahim, Wan Rozita Wan Mahiyuddin

**Affiliations:** 1Department of Community Health, Universiti Putra Malaysia, Serdang 43400, Malaysia; 2Department of Community Health, Universiti Kebangsaan Malaysia, Cheras 56000, Malaysia; 3Faculty of Health Sciences, Universiti Teknologi MARA, Bandar Puncak Alam 42300, Malaysia; 4Public Health Medicine Department, Faculty of Medicine and Health Sciences, Universiti Malaysia Sabah, Kota Kinabalu 88400, Malaysia; 5Institute for Medical Research, National Institute of Health, Ministry of Health, Shah Alam 40170, Malaysia

**Keywords:** depression, anxiety, stress, intervention

## Abstract

Depression, anxiety, and stress (DAS) among adolescents have become a public health concern. The aim of this study was to develop, implement, and measure an IMB-based health education intervention module for reducing DAS among adolescents in boarding schools in the state of Negeri Sembilan, Malaysia. A single-blinded cluster randomised control trial (RCT) was conducted among students with abnormal DASS-21 scores. They were divided into an intervention group (three schools, 62 participants) and a control group (three schools, 57 participants). Participants in the intervention group received IMB-based health education, while participants in the control group underwent the standard care session. To determine the effectiveness of the intervention, the Generalised Linear Mixed Model (GLMM) analysis was conducted. A total of 119 students participated in this study, and no loss to follow-up was reported. Both intervention and control groups showed significantly reduced DAS scores (*p* < 0.005). However, the reduction of these scores was greater in the intervention group. The GLMM analysis revealed that the intervention was effective in reducing depression (ß = −2.400, t = −3.102, SE = 0.7735, *p* = 0.002, 95% CI = −3.921, −0.878), anxiety (ß = −2.129, t = −2.824, SE = 0.7541, *p* = 0.005, 95% CI = −3.612, −0.646), and stress (ß = −1.335, t = −2.457, SE = 0.536, *p* = 0.015, 95% CI = −2.045, −0.266) among adolescents. The IMB-based health education module was effective in reducing DAS among adolescents in boarding schools.

## 1. Introduction

Globally, adolescents have been reported to have a high prevalence of mental health disorders [1,2]. In 2021, the World Health Organization (WHO) projected that there were 1.2 billion adolescents among the world population, and one in seven (10–19 years old) has mental health problems [3]. The available data showed that 13% of adolescents lived with diagnosed mental health disorders, and this percentage represents 86 million adolescents aged 15–19 and 89 million adolescents aged 10–14 [3]. Additionally, 40% of the diagnosed mental health disorders among adolescents were depression and anxiety [4]. The prevalence of depression, anxiety, and stress (DAS) among adolescents in low- and middle-income countries was similar to the prevalence of DAS in high-income countries [5].

The inadequate response to this problem has caused adolescents to suffer, face disability, and even death, as well as interfering with their education and ability to reach their full potential. To make it worse, the recent COVID-19 pandemic may worsen the depression, stress and anxiety conditions among adolescents. The United Nations International Children’s Emergency Fund (UNICEF) estimated that 45,000 adolescents die each year (one in every 11 min) due to mental health disorders. This number amounts to an annual human capital loss of USD 387.2 billion, whereby 92% (USD 340.2 billion) are due to depression and anxiety [4]. Despite this significant loss, governments in some of the poorest countries in the world have spent less than 1% of their health budget to treat mental health disorders, and even less than 2% in the lower- and middle-income countries [6]. The mental health budget in Malaysia accounts for approximately 0.28–0.39% of the total health budget [7]. This expenditure is grossly disproportionate to the burden of mental health disorders, and projects a very low value on human psychological and mental well-being. 

The 2017 National Health and Morbidity Survey (NHMS) has shown an alarming picture of the mental health state of Malaysian adolescents, whereby the prevalence of depression, anxiety, and stress among secondary school students was 18.3%, 39.7%, and 9.6%, respectively [8]. The prevalence of depression was the highest among males, while anxiety was the highest among females [9]. Moreover, depression and anxiety disorders accounted for 2.26% and 1.71% of the total disability-adjusted life years (DALYs), respectively, and mental health disorders have affected Malaysia’s economy by MYR 14.46 billion in 2018 [10]. 

Even though students might suffer from being apart from the family, and experience higher levels of anxiety and stress, most parents believe that boarding school is the only way to ensure their children receive quality education and confirm their place in the top higher education institutes [11]. Studies on the prevalence of DAS among boarding school students in Malaysia are limited. Nonetheless, a previous study reported that the burden of DAS among adolescents in Malaysian boarding schools was higher compared to the NHMS findings, whereby the prevalence of depression, anxiety, and stress was at 39.7%, 67.1%, and 44.9%, respectively [12]. Another study in northeast Peninsula Malaysia reported that the prevalence of stress in boarding school was 26.6% [13]. These scenarios may deteriorate further, as the majority of adolescents in secondary schools never use the health care services offered by the Ministry of Health, since they often seek assistance from informal sources [14].

Several types of interventions to reduce DAS are practised in other countries, such as transcendental meditation, learning to breathe, stress management programme, cognitive behavioural therapy, relaxation programme, mindfulness stress reduction programme, and health education intervention [15]. In Malaysia, despite the implementation of school-based mental health interventions, studies were still reporting a high prevalence of DAS among adolescents [16,17]. Thus, the main objective of this current study was to develop a health education module based on the Information–Motivation–Behavioural Skills (IMB) theoretical model, and subsequently, implement and evaluate its effectiveness in reducing DAS among adolescents in boarding schools. Despite the issues mentioned earlier, no randomised control trial (RCT) study has been conducted to measure the effectiveness of a theory-based health education intervention module for reducing DAS among adolescents, specifically in boarding schools. Hence, the results of this study can contribute to the body of knowledge with an effective, feasible, appropriate, and convenient school-based intervention to reduce DAS among adolescents in boarding schools.

## 2. Materials and Methods

This study was conducted in boarding schools in the state of Negeri Sembilan, Malaysia, during the COVID-19 pandemic. It was a parallel single-blinded RCT, with pre- and post-intervention follow-ups. DAS scores were measured at the one-month and two-month follow-ups. Originally, seven boarding schools were involved; however, one school was declared as a COVID-19 cluster, and thus, they withdrew their participation. The remaining six boarding schools were randomly assigned into an intervention group and a control group. The random assignment was conducted by preparing six 34 × 24 cm envelopes, and each envelope contained a 30 × 21 cm card labelled either ‘A’ for intervention, or ‘B’ for control. These envelopes were lined on the inside with carbon paper to serve as a trail for audit purposes. The school counsellors participated in the randomisation procedure by choosing one envelope to represent their school. They were then instructed to write the name of their school on the front of the randomly picked envelope and to hand it over to the investigator promptly. The contents and information included in the envelope were unknown to the counsellors. Random allocation of the clusters was made by tracking the name of the school on the selected envelope and on the card.

Students from each school were selected using the simple random technique from the list of eligible students provided by the school counsellors. Since each school has an equal number of students, each school has an equal number of participants. The allocation of schools and students into the intervention and control groups were unknown to all participants, since they were blinded during every occasion. The participants in the intervention group received the theory-based health education module, while the participants in the control group received the standard care sessions. The clusters and participants were blinded from the beginning until the end of this study. These participants were selected based on predefined inclusion and exclusion criteria. The inclusion criteria included participants who have the consent of their parents, are able to understand and write in Malay and English, and have been screened using DASS-21 with abnormal finding. On the other hand, students with severe, or extremely severe depression, and/or anxiety, and/or stress were excluded from this study. Students who did not obtain permission from the school principal or under medical/psychiatric follow-up were also excluded from this study. In total, 124 students were selected at the beginning of the study. However, five students from the control group were not interested in participating. Subsequently, 121 students had their follow-up assessment through the data collection period. No loss to follow-up was reported in this study. Figure 1 shows the flow chart of the study.

### 2.1. IMB-Based Health Education Module

The information–motivation–behavioural skills (IMB)-based health education module was developed in the Malay language, and its content was validated by several experts (public health medicine specialist, family medicine specialist, psychiatrist, and specialist in health education). This module was also tested in a pilot study to assess its feasibility and acceptability among 10 participants. Their ideas and comments on the module were taken into account. Additionally, this module was designed to fit the non-medical personnel use as much as possible, such as teachers and school counsellors for the purpose of health education intervention at schools.

This module was adapted from several available sources, such as the Engaging the Adolescent Module Using HEADSS Framework by the Ministry of Health (MOH), the Mental Health Handbook by the Malaysian Mental Health Association, the Malaysian Dietary Guidelines for Children and Adolescents by MOH, the Healthy Mind Module (first and second edition) by MOH, How to Manage and Reduce Stress by the United Kingdom Mental Health Association, and Doing What Matters in Times of Stress: An Illustrated Guide by the World Health Organization (WHO) [18,19,20,21,22,23]. The information in these sources has been summarised and rewritten by the investigator. Furthermore, the activities and programmes in this module were developed in accordance with the determinants of the IMB theoretical model (i.e., information, motivation, and behavioural skills). For example, the information determinants consisted of two health talks with slide presentations that have been designed specifically for students in boarding schools. The main objective was to enhance their knowledge on mental health aspects, namely, depression, anxiety, and stress. On the other hand, group discussion activity, with sharing experience and coping strategies, was intended to motivate the students and help them develop a positive attitude towards DAS, as well as to improve their self-efficacy. Next, the objective of the 10 Bs in the behavioural skills determinant was to demonstrate and practice mental health coping techniques, such as breathing technique, prayer recitation, and massage technique. Apart from the theory-based health education, this intervention module was also integrated with spiritual approaches. The material was written with clear communication principles, such as clear words, bullet points, colour coding, illustrations, and step-by-step directions. 

### 2.2. Implementation of the IMB-Based Health Education Module

There were four sessions in total, whereby two sessions for information determinant and one session each for motivation and behavioural skills determinants. These group-based intervention sessions lasted 90 min each and were conducted once per week for one month. The IMB-based health education intervention module was delivered via different approaches, such as interactive sessions, health talk with slide presentation, handouts, question and answer sessions, group discussion, counselling, video presentation, and instruction and demonstration. The information determinant was conducted in two sessions, with the first session starting with an introductory activity. The intention of this activity was to reveal the purpose of the meeting, gain the attention of the participants, preview the study, and establish trust and credibility. The investigator introduced himself, greeted all the participants, and explained the purpose and objective of the meeting. The participants were then given the opportunity to introduce themselves. During the second 90 min session, to enhance their knowledge, the participants were taught mental health risk factors and strategies to overcome problems. Then, a group discussion for sharing experiences and coping strategies was held during the third intervention session for the motivation determinant. The investigator shared his own life experiences and how he faced problems. Then, volunteers among the participants were invited to share their experiences and problems. All findings on coping strategies were listed and discussed. The final session was on the behavioural skills determinant, whereby participants were given a demonstration on mental health coping strategies, which they later practiced. The 10 Bs and breathing massage techniques were demonstrated during this session. Participants were also taught prayer recitation that can be practiced whenever they are facing problems. 

To fit with the COVID-19 constrain and conditions imposed by the Ministry of Education (MOE), the module was presented virtually via Zoom meeting instead of face to face. A WhatsApp group was created for each school to facilitate communication between the investigator and participants. Scheduled Zoom meeting links for each intervention session were shared with the participants via WhatsApp. Participants were reminded about each meeting three times; three days before, one day before, and one hour before. 

### 2.3. Data Collection Process

Data collection was conducted over a six-month duration from 1 April until 30 September 2021. Due to the prevalence of COVID-19 and the conditions imposed by the MOE, the investigator was not permitted to enter school premises and meet the students. This study has to be conducted online to minimise COVID-19 infection and outbreak in these boarding schools by limiting physical and social interactions between students and outsiders. Discussions and meetings with school counsellors were also conducted virtually. Once ethics clearance and authorisation were obtained, the investigator took the initiative to contact all school counsellors and conducted an online meeting via Zoom meeting. They were explained about the study and data collection process. The school counsellors were mandated to give an early briefing about this study to the selected students. Next, separate online meetings via Zoom meeting were conducted for all participants from each school. Participants were evaluated and checked to determine whether they were eligible for this study based on the inclusion and exclusion criteria. Since the participants were at school, their counsellors facilitated the online meetings by gathering them in an appropriate place, such as the school hall, a special classroom, or the counselling room. 

The investigator explained the study to the participants, and they were informed that participation in this study was voluntary and parental consent was required. During this study, the participants were allowed to withdraw and terminate his/her participation at any time. Participants who missed a follow-up, changed their mind about participating, or withdrew their participation were terminated from this study immediately. They were also briefed on what they have to do, the benefits of this study, and information confidentiality. The participants were given sufficient time to consider their participation in this study. Students who verbally expressed their agreement to participate were given the respondent’s information sheet (RIS) and participant consent form. The RIS and parent/guardian consent forms were physically mailed to the participants’ home address, and softcopies of these forms were also emailed to their parents. The participants’ consent forms were collected by school counsellors and forwarded to the investigator, while the parent/guardian consent forms were directly mailed to the investigator.

For the baseline study, a set of questionnaire booklets was physically mailed to each school. Each envelope containing the booklets was addressed to the school counsellor, and ‘confidential’ was written at the top of the envelope. The same date and time were scheduled for all participants at each participating school to answer the questionnaire, apart from the teaching and learning time at school (PnP). For this purpose, the participants were gathered in a comfortable place (e.g., classroom, hall, or counselling room) and the questionnaire booklets were distributed. All participants were given enough time to answer all the questions. Upon completing the questionnaire, the booklets were collected by the school counsellors and mailed to the investigator. 

Before the commencement of the intervention, the MOE declared closure of all schools in the country and implemented online teaching and learning at home (PdPR). Thus, the intervention was completely delivered online. Meanwhile, data from the follow-up study at one-month and two-month post intervention sessions were collected via Google Form. The questionnaire was transformed into an online set with mandatory questions (participants must answer all questions in order to submit their response). Prior to that, WhatsApp groups were created for each school to facilitate communication between the investigator and participants. Google Form links were attached in their respective WhatsApp group for the one-month and two-month post-intervention sessions. In addition, all participants were reminded to complete their follow-ups. Finally, all participants were thanked for their time and received tokens of appreciation. Following the completion of the follow-up study, the WhatsApp groups were deactivated.

The total duration for data collection process for this study was six months. Although the intervention and data collection were conducted simultaneously among participants from one school, due to researcher limitation, the intervention and data collection were not able to be collected from all three schools at the same time. 

### 2.4. Study Instruments

This study used a validated self-administered questionnaire, both English and Malay versions. The sociodemographic section was initially developed in English, which consisted of several items, such as age, education level (form), gender, ethnicity, average monthly household income, father’s and mother’s highest education levels, and underlying medical history. The classification of household income was based on the Household Income and Basic Amenities Survey Report 2019, Department of Statistics Malaysia [24]. Except for age, all questions were in the close-ended form. 

DASS-21 was developed by Lovibond and Lovibond in 1995, which has been widely use in clinical samples to screen for depression, anxiety, and stress [25]. In other words, it was not intended to be a diagnostic questionnaire. DASS-21 consisted of three self-report scales designed to measure negative emotional states, namely, depression, anxiety, and stress [26]. For each item, participants are required to rate the extent to which they have experienced the given state over the past week. In this study, the validated DASS-21, Malay version, with high psychometric properties were adapted from Musa at el. (2007), with Cronbach’s alpha of 0.84 for depression, 0.74 for anxiety, and 0.79 for stress [27,28]. Participants were asked to rate each statement to indicate their feeling for the past one week based on the following rating scales: 0 = did not apply to me at all (never), 1 = applied to me to some degree, or some of the time (sometimes), 2 = applied to me to a considerable degree, or a good part of the time (often), and 3 = applied to me very much, or most of the time (always) [29].

### 2.5. Quality Assurance

The process of translating the English questionnaire into Malay was conducted in two phases: (i) Phase 1: translation, expert panel endorsement, and back translation; and (ii) Phase 2: cultural validation [30]. A forward translation was performed by a native speaker of the Malay language, who was also fluent in the English language. This was followed by a back-translation, which involved a panel of experts (public health physicians and English teacher). The translated questionnaire was then pre-tested among participating students in a boarding school. Participants’ comments were reviewed and applied; consequently, the final version of the questionnaire was completed. The content validity of the questionnaire was assessed in a collaboration with four mental health experts (public medicine health specialist, family medicine specialist, psychiatrist, and health education specialist) to determine the appropriateness and magnitude in which the constructs of interest were represented by the items in the questionnaire. The face validation of the questionnaire was conducted among 10 randomly selected students. School counsellors were also given the opportunity to weigh in on the appropriateness of the questionnaire. It was pre-tested to determine its reliability before data collection was conducted among a convenient sample of 30 students in the selected school. The Cronbach’s alpha values of the knowledge section on mental health and DAS, and self-efficacy were more than 0.8. Meanwhile, the Cronbach’s alpha value of the attitudes section towards DAS was more than 0.6, which was acceptable in this study [31].

### 2.6. Data Analysis

Data were compiled and analysed using the Statistical Package for Social Sciences (IBM SPSS, version 26.0, Armonk, NY, USA). The significance level was accepted at 0.05, and the confidence interval was at 95%. Since the collected data were normally distributed, the comparison between the intervention and control groups was made using the Chi-square test at baseline. Then, comparisons were made between these groups using the Mann–Whitney U test (not normally distributed data), and within each group using the ANOVA repeated measures during follow-ups. A multi-level modelling analysis using the Generalised Linear Mixed Model (GLMM) was conducted to determine the interaction effects of the IMB-based health education module on depression, anxiety, and stress scores. There were no missing data in this study; therefore, per-protocol analysis was applied to measure the effect of actually receiving the IMB-based health education intervention module throughout the whole follow-up period. 

### 2.7. Ethical Approval

This study obtained the ethical clearance from the Ethics Committee for Research Involving Humans, Universiti Putra Malaysia (UPM) (Ref. no. JKEUPM-2020-511). In addition, the permission to execute this study in boarding schools was approved by the Ministry of Education, Malaysia (MOE). This study was also registered with the Thai Clinical Trial Registry (Ref. no. TCTR20220625002).

## 3. Results

### 3.1. Baseline Participant Characteristics

Out of 3600 students, 345 Form 1, Form 2, and Form 4 students were found eligible for this study (168 participants from the intervention group and 177 participants from the control group). To execute this study, 124 students were recruited, as both groups were allocated with 62 participants. However, five students from the control group were not interested in participating in this study. In total, 119 participants underwent a baseline assessment, 62 participants received intervention, and 57 participants received the standard care session (control group). The response rate in this study was 95.9%.

The age of the participants ranged between 13 and 16 years old, and the majority were Form 4 students (*n* = 47, 39.5%). All of them were Malay, and 61.2% (*n* = 74) of the participants were female. Most participants (*n* = 44, 37%) came from a family with household income of MYR 4850 to 10,959 (M40 income group), followed by MYR 4849 and below (B40 income group), and MYR 10,960 and above (T20 income group). In terms of their father’s and mother’s education level, all participants reported that their parents received formal, or primary schooling. The majority of the participants’ father have STPM/Diploma (*n* = 38, 31.9%). On the other hand, most of the participants’ mothers obtained a higher education level (Bachelor’s degree) (*n* = 40, 33.6%).

Participants in the intervention group were mostly male (*n* = 34, 54.8%), from a family with a higher household income (M40 income group, *n* = 28, 45.2%), and their parents both have higher education levels compared to participants in the control group. On the other hand, the majority of participants in the control group were female (*n* = 45, 78.9%), from a family with a lower household income (B40 income group) (*n* = 29, 50.9%), and both parents with lower education level. The majority of the participants reported that they have no medical illness (*n* = 101, 81.5%). In the intervention group, only 11 participants (17.7%) reported that they have current underlying medical illnesses, and seven participants (12.35) in the control group reported the same. There was no significant difference regarding medical illnesses between the intervention and control groups (χ^2^ = 0.690, *p* = 0.406) at baseline.

Table 1 shows the distribution of the sociodemographic characteristics between the intervention and control groups. Significant differences were found in the participants’ gender, household income, father’s education level, and mother’s education level (*p* > 0.05) between the intervention and control groups. These significant factors were controlled in the mixed model. In addition, DAS scores between the intervention and control groups showed no significant difference (*p* > 0.05) (refer Table 2).

### 3.2. Comparisons of Depression, Anxiety, and Stress Scores between Groups and within Each Group during Follow-Up Sessions

Depression, anxiety, and stress scores showed significant differences between the intervention and control groups (*p* < 0.05) (refer Table 3). The depression scores over repeated measurements showed a significant difference within the intervention group (F(1.011, 61.684), *p* < 0.001) = 81.259, η^2^ = 0.571). The post hoc test showed significant differences between baseline and one-month scores, and between baseline and two-month scores. A significant difference of depression scores was also observed within the control group (F(1.027, 57.507) = 14.975, *p* < 0.001, η^2^ = 0.211). The post hoc test showed significant differences between baseline and one-month scores, and between baseline and two-month scores. A significant difference in anxiety scores over repeated measurements was also found within the intervention group (F(1.039, 63.362) = 193.629, *p* < 0.001, η^2^ = 0.760). The post hoc test showed significant differences between baseline and one-month scores, and between baseline and two-month scores, as well as between one-month and two-month scores. A significant difference in the anxiety scores was also observed within the control group (F(1.090, 61.043) = 14.424, *p* < 0.001, η^2^ = 0.205). The post hoc test showed significant differences between baseline and one-month scores, and between baseline and two-month scores. Significant differences were observed in stress scores across the study period within the intervention group (F(1.56, 64.411) = 715.549, *p* < 0.001, η^2^ = 0.921) and within the control group (F(1.243, 69.629) = 33.034, *p* < 0.001, η^2^ = 0.371). The post hoc test for the intervention group showed significant differences between baseline and one-month scores, and between baseline and two-month scores. The post hoc test for the control group showed similar findings. DAS score comparisons within each group are summarised in Table 4.

### 3.3. Effect of Intervention on Depression, Anxiety, and Stress Scores

The depression score of a participant in the intervention group was 2.4 points lower than a participant in the control group (ß = −2.400, t = −3.102, SE = 0.7735, *p* = 0.002, 95% CI = −3.921, −0.878). A significant difference in depression score was also observed at the two-month follow-up, whereby the participants obtained 13.86% lower depression scores compared to the baseline. Participants in the intervention group scored 3.991 and 3.985 points lower at the one-month and two-month follow-ups, respectively (refer Table 5). 

More participants in the intervention group significantly reduced their anxiety level (lower anxiety score) (ß = −2.129, t = −2.824, SE = 0.7541, *p* = 0.005, 95% CI = −3.612, −0.646) compared to participants in the control group. The anxiety score of a participant who received the health education intervention was 2.129 points lower than the participant in the control group. Significant differences were observed in the anxiety scores of participants in the intervention group, which reduced by 21.40% and 19.65% at the two-month and one-month follow-ups, respectively, compared to the baseline. Participants in the intervention group scored 4.478 and 4.714 points lower at the one-month and two-month follow-ups, respectively (refer Table 5). 

The results also showed that more participants in the intervention group than in the control group reduced their stress level (lower stress score) (ß = −1.335, t = −2.457, SE = 0.536, *p* = 0.015, 95% CI = −2.045, −0.266). Thus, the stress score of a participant in the intervention group was significantly lower by 1.335 points compared to a participant in the control group. Additionally, the stress scores of the intervention group were reduced by 18.25% and 16.67% at the two-month and one-month follow-ups, respectively, compared to the baseline. Participants in the intervention group scored 7.817 and 7.853 points lower at the one-month and two-month follow-ups, respectively (refer to Table 5). 

## 4. Discussion

Comparisons between groups at the one-month and two-month follow-ups and within group exhibited significant differences of the depression score. The depression scores were lower among participants who received intervention than those who did not. Further analysis confirmed that the intervention group showed a considerably greater reduction in depression scores than the control group. In low-middle income countries, multicomponent interventions consisting of information materials and educational intervention have been proven effective to lower depression among adolescents [32,33]. In China, health education intervention was also reported to significantly lower anxiety and depression levels among adolescents during the COVID-19 pandemic, and it improved their sleep quality at the same time [34]. In contrast, a study in Southwestern America in 2015 reported that health promotion at schools has no significant effect on depression and anxiety at the immediate and six-month post-intervention follow-up [35]. Negative findings were also reported in Australia, whereby instead of a reduction in smoking among students, health education intervention showed no significant effect on students’ depressive symptoms [36].

Within-group comparisons revealed a substantial decrease in the anxiety scores of both intervention and control groups. These comparisons also showed that the reduction of anxiety scores of the intervention group was greater compared to the anxiety scores of the control group. Apart from that, when the two groups were compared using GLMM, the intervention group experienced a significantly higher reduction of anxiety scores. These findings on reduced anxiety levels among adolescents following health education intervention were supported by previous international RCT studies [37,38,39]. The findings of this study were consistent with the most similar parallel cluster RCT study in Malaysia, which analysed the effect of health education on reducing anxiety among school-aged students at immediately post-intervention and three-month follow-up [40]. Their study proved that a relatively short health education intervention period (four weeks) without a booster programme was significantly effective in reducing anxiety scores among students. Apart from that, religion in general (i.e., religious training, spirituality, faith, and prayer) was also associated with reduced depression, anxiety, and stress levels [41,42,43]. It has been proven that Islamic teaching, integrated into a health education intervention module, has a considerable beneficial effect on anxiety among female adolescents in Iran, at both immediate and three-month follow-ups [44]. An RCT study was conducted to determine the effect of a health education intervention, known as the Aussie Optimism Programme, on depression and anxiety among adolescents from disadvantaged schools in Australia. This programme consisted of social life and thinking skills that have been revealed to reduce the mean score of depression and anxiety among participants over time. However, this intervention has no significant effect on the participants [45]. The lack of effect on depression and anxiety levels was reportedly due to the low pre-test level of students’ internalised symptoms despite their low socioeconomic status. Furthermore, their study had a long follow-up period of six to 18 months, which could have caused the students to forget and ignore the lessons. Similar findings were observed in another study in Canada, whereby health education via the FRIENDS for Life Programme was reported to be ineffective in reducing anxiety among the total sample of Aboriginal children [46].

On the other hand, the IMB-based health education intervention module was effective in reducing stress among adolescents in boarding schools in Negeri Sembilan. At the one-month and two-month follow-ups, the participants showed a substantial reduction in stress scores within their own group. The stress mean score of the intervention group was significantly lower than the mean score of the control group. Apart from that, when the performance of both groups was compared using GLMM, the reduction in stress levels among participants who received the intervention was significantly remarkable. This finding was supported by a similar RCT study in Malaysia on improving emotional health and self-esteem among adolescents. The health education programme applied in this study has significantly reduced the level of emotional problems (depression, anxiety, and stress), and the intervention was reported to have the greatest impact on stress [47]. Another study similarly reported that health education has a positive impact on adolescent stress level [48]. This eight-week stress management programme consisted of a healthy lifestyle, exercises, breathing technique, muscle relaxation, healthy diet, and relaxation techniques, which showed a significant reduction in anxiety and compulsive behaviour, as well as increased resilience and self-evaluation of school performance among participants. 

### 4.1. Strength and Limitation 

This RCT study provided a high level of evidence. The randomisation step at the school level minimised cross-contamination between the intervention and control groups. The response rate was high, and the sample size was adequate. In addition, the collected data were analysed using a proper statistical analysis, since the significant findings at baseline were controlled in the GLMM analysis. Apart from that, this theory-based intervention was prepared in the Malay language (Malaysia’s national language) to ensure comprehension and acceptability by the participants, as well as its feasibility for delivery. This intervention was also concise, simple, and required minimal cost. Nonetheless, there were some limitations faced by this study, since it was conducted during the COVID-19 pandemic. The uncertain atmosphere led to prolonged school closure and execution of online teaching and learning at home. Consequently, this research and the intervention sessions had to be conducted online. Another limitation was the use of online self-administered questionnaire during follow-ups, which could have led to some bias, such as misunderstanding of the questionnaires. However, providing details and explanations to the participants prior to data collection could have minimised this limitation. In addition, this study was a single-blinded RCT, and the investigator personally delivered the health education module to the intervention group, which could be attributed to the experimenter effect. 

### 4.2. Recommendation 

This theory-based health education module can be implemented in boarding schools to reduce depression, anxiety and stress among the students, provided that the schoolteachers or counsellors are trained. Apart from acting as an intervention, it can also serve as a health promotion which can be embedded in the school curriculum and co-curriculum. 

We also recommend for collaboration with other stake holders such as the Ministry of Youth and Sport, and Ministry of Women, Family and Community Development and non-governmental organizations (NGOs) by embedding this module into the current programs and activities which are focusing on adolescent mental health at the community and institutional levels. 

This study has proven that the intervention was effective in reducing depression, anxiety, and stress among adolescents at one-month and two-month follow-up. The effectiveness of a health-education module could be measured at a longer follow-up period (e.g., 4 or 6 months) to look for its sustainability in reducing adolescents’ depression, anxiety and stress. Researcher may also think of administering booster intervention and look for its effect if there are any changes in depression, anxiety and stress over time. Furthermore, future studies could also consider the use of any mechanism to document the participants’ compliance to the intervention module (e.g., diary). For example, the participant may document the frequency of physical activity, types and amount of food they take, daily activities at home and school and frequency of practising the strategies to overcome depression, anxiety and stress. 

Since the health education intervention module in this study was delivered as an online intervention, we strongly recommend for other researchers to replicate the study as a face-to-face intervention in future research. The question remains of whether online-based intervention is equally as beneficial for adolescents as standard face-to-face intervention. It is important to compare the mode of the intervention (online vs. face-to-face) and observe for any different outcomes. In addition, if the findings are comparable, we should look for its feasibility, acceptability, adherence of the participants, duration, time of the implementation (during school day or holiday) and cost of the implementation of the intervention module. 

## 5. Conclusions

Data analysis proved that the IMB-based health education intervention module was significantly effective in reducing depression, anxiety, and stress levels among adolescents in boarding schools in the state of Negeri Sembilan. This study observed significant reductions of depression, anxiety, and stress scores within the intervention group, and between the intervention and control groups at follow-up time points. Although the control group showed similar results, the magnitude of reduction in the depression, anxiety, and stress scores of the intervention group was greater. 

## Figures and Tables

**Figure 1 ijerph-19-15362-f001:**
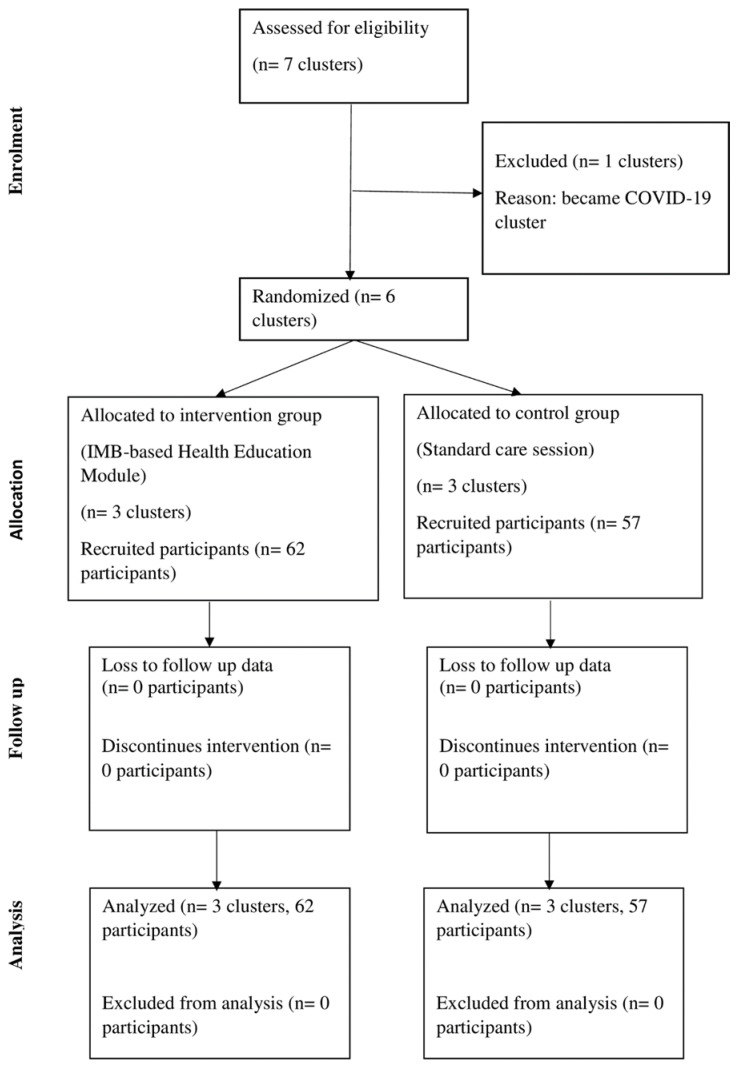
Flow chart of the study.

**Table 1 ijerph-19-15362-t001:** Baseline sociodemographic characteristics.

Characteristics	Intervention		Control			
	Frequency *n* = 62	%	Frequency *n* = 57	%	χ² (df)	*p* Value
**Education Level**					2.544 (2)	0.280
Form 1	18	29.0	17	29.8		
Form 2	23	37.1	14	24.6		
Form 4	21	33.9	26	45.6		
**Gender**					14.296 (1)	0.001 *
Male	34	54.8	12	21.1		
Female	28	45.2	45	78.9		
**Ethnicity**						
Malay	62	100	57	100		
Others	0	0	0	0		
**Household income (RM)**					10.313 (2)	0.006 *
4849 and below	14	22.6	29	50.9		
4850–10,959	28	45.2	16	28.1		
10,960 and above	20	32.3	12	21.1		
**Father’s education level**					9.311 (3)	0.025 *
No formal education	0	0	0	0		
Primary school	0	0	0	0		
PMR/SPM	8	12.9	20	35.1		
STPM/Diploma	20	32.3	18	31.6		
Bachelor’s degree	22	35.5	12	21.1		
Master’s degree/ PhD	12	19.4	7	12.3		
**Mother’s education level**					9.011 (3)	0.029 *
No formal education	0	0	0	0		
Primary school	0	0	0	0		
PMR/SPM	10	16.1	22	38.6		
STPM/Diploma	20	32.3	18	31.6		
Bachelor’s degree	26	41.9	14	24.6		
Master’s degree/	6	9.7	3	5.3		
PhD						
**Health Problem**					0.690 (1)	0.406
Yes	11	17.7	7	12.3		
No	51	82.3	50	87.7		

* Significant at *p* < 0.05. PMR/SPM, lower secondary assessment/Malaysian education certificate. STPM, Malaysian higher education certificate.

**Table 2 ijerph-19-15362-t002:** Baseline comparison of depression, anxiety, and stress scores.

DAS	Mean (SD)		df	*t*-Value	Mean Diff. (95% CI)	*p* Value
	Intervention	Control				
**Depression**	5.58 (5.02)	7.00 (6.10)	117	1.389	1.419 (−0.605, 3.443)	0.168
**Anxiety**	7.02 (38.26)	7.98 (63.17)	90.671	0.999	0.966 (−0.956, 2.888)	0.321
**Stress**	9.89 (3.06)	10.39 (4.10)	103.182	0.746	0.499 (−0.827, 1.825)	0.457

Significant at *p* < 0.05.

**Table 3 ijerph-19-15362-t003:** Post-intervention comparisons of depression, anxiety, and stress scores between groups.

DAS	Group Median (IQR)			
	Intervention	Control	Mann-Whitney U Test	*p* Value
**Depression**				
one-month	0 (0)	5 (7)	2115.000	<0.001 *
two-month	0 (0)	5 (7)	184.000	<0.001 *
**Anxiety**				
one-month	1 (1)	5 (6)	172.000	<0.001 *
two-month	0 (0)	5 (6)	87.000	<0.001 *
**Stress**				
one-month	0 (0)	5 (7)	2115.000	<0.001 *
two-month	0 (0)	5 (7)	184.000	<0.001 *

* Significant at *p* < 0.05.

**Table 4 ijerph-19-15362-t004:** Post-intervention comparisons of depression, anxiety, and stress scores within each group.

	Type III Sum of Squares	df	F	*p* Value	Partial ETA η^2^
**Depression**					
Within intervention group	1178.204	1.011, 61.684	81.259	<0.001 *	0.571
Within control group	69.485	1.027, 57.507	14.975	<0.001 *	0.211
**Anxiety**					
Within intervention group	1834.677	1.039, 63.362	193.629	<0.001 *	0.760
Within control group	160.982	1.090, 61.043	14.424	<0.001 *	0.205
**Stress**					
Within intervention group	3795.097	1.056, 64.411	715.459	<0.001 *	0.921
Within control group	111.503	1.243, 69.629	33.034	<0.001 *	0.371

* Significant at *p* < 0.05.

**Table 5 ijerph-19-15362-t005:** The effect of IMB-based health education on depression, anxiety, and stress scores between the intervention and control groups.

Variable	Coefficients	Std. Error	t	*p* Value	95% CI	
					Lower	Upper
**DEPRESSION**						
** *Group* **						
Intervention	−2.400	0.7735	−3.102	0.002 *	−3.921	−0.878
Control	1					
** *Time* **						
Two-month follow-up	−1.386	0.6858	−2.021	0.044 *	−2.735	−0.037
One-month follow-up	−1.316	0.6858	−1.919	0.056	−2.665	0.033
Baseline	1					
** *Time*Group* **						
Two-month*Intervention	−3.985	0.9501	−4.195	<0.001 *	−5.854	−2.116
One-month*Intervention	−3.991	0.9501	−4.2	<0.001 *	−5.859	−2.122
Baseline*Intervention	1					
**ANXIETY**						
** *Group* **						
Intervention	−2.129	0.7541	−2.824	0.005 *	−3.612	−0.646
Control	1					
** *Time* **						
Two-month follow-up	−2.140	0.6146	−3.483	0.001 *	−3.349	−0.932
One-month follow-up	−1.965	0.6146	−3.197	0.002 *	−3.174	−0.756
Baseline	1					
** *Time*Group* **						
Two-month*Intervention	−4.714	0.8515	−5.537	<0.001 *	−6.389	−3.040
One-month*Intervention	−4.487	0.8515	−5.269	<0.001 *	−6.161	−2.812
Baseline*Intervention	1					
**STRESS**						
** *Group* **						
Intervention	−1.335	0.536	−2.457	0.015 *	−2.045	−0.266
Control	1					
** *Time* **						
Two-month follow-up	−1.825	0.4714	−3.871	<0.001 *	−2.752	−0.897
One-month follow-up	−1.667	0.4714	−3.536	<0.001 *	−2.594	−0.74
Baseline	1					
** *Time*Group* **						
Two-month*Intervention	−7.853	0.653	−12.025	<0.001 *	−9.137	−5.668
One-month*Intervention	−7.817	0.653	−11.971	<0.001 *	−9.102	−6.533
Baseline*Intervention	1					

* Significant at *p* < 0.05.

## Data Availability

Data are available upon direct request to the corresponding author, Rahmat Dapari, drrahmat@upm.edu.my, subject to approval by the supervisory committee.

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
