# Peer review of "Health Education Module Based on Information–Motivation–Behavioural Skills (IMB) for Reducing Depression, Anxiety, and Stress among Adolescents in Boarding Schools: A Clustered Randomised Controlled Trial"

_ijerph, 2022, doi:10.3390/ijerph192215362_

Round 1

Reviewer 1 Report

The manuscript ijerph-2014252 "Health Education Module Based on Information-Motivation- Behavioural Skills (IMB) for Reducing Depression, Anxiety, and Stress Among Adolescents in Boarding Schools: A Clustered Randomised Controlled Trial" is a case-control study that analyzes the effectiveness of a mental health intervention in reducing the level of stress, anxiety, and depression among boarding school students.   

The manuscript is clear, brief, precise, relevant and novel. The objective is clearly and precisely stated and the results and discussion are developed in a structured way with the proposed objectives. In the results, it is necessary to adapt the information contained in the tables to facilitate comparison and reading. The discussion is presented in an orderly and well-integrated manner. 

It is a consistent and correctly developed manuscript, being, for the most part, easy to read. 

Nevertheless, a major revision of the present version before publication is recommended in order to include some suggestions for improvement, which mainly affect the clarification of the information contained in the tables of results. 

Specific remarks:

1- Introduction:

The wording is concise and in the right direction, from the concrete to the particular, all the necessary terms being defined. The research question is well stated. 

It would be convenient to include in the state of the question the data, if known, of adolescents with problems of depression, anxiety or stress during the pandemic, which complements the information provided on the prevalence of DAS included in the L47-58.

It would also be enriching to indicate whether within the curriculum there is any subject in the schools in which the contents addressed in the intervention are addressed. 

2. Materials and methods: 

The material and methods are precisely described, as are the inclusion and exclusion criteria. 

2.3 Data collection process

Although this section of the manuscript reflects all the expected information, its reading is complex and extensive, making it difficult to follow the section. Although it is not usual in material and methods, it is recommended that a figure (diagram) be included to clarify the text or the rewrite so that it is easy to read.

in this section it is also recommended to include how data collection is carried out on all participants at the same time (information included in later sections). 

2.4 Study instruments: 

L269-L272: "They were advised not to spend too much time on each statement. In addition, their knowledge on mental health and DAS, their attitude towards DAS, and self- 270 efficacy were assessed using another set of questionnaires adapted from different sources 271 (31-33)." 

The name of the questionnaires referenced in the text must be included (the bibliographic reference is not sufficient). 

3. Results:

The results in the text are adequately reflected.  Improvements in the presentation of the tables are recommended.  

Table 2 Initial comparison of depression, anxiety and stress scores; Table 3 Post-intervention comparisons of depression, anxiety and stress scores between groups; and Table 4 Post-intervention comparisons of depression, anxiety and stress scores within each group can be integrated into a single table. 

If it is desired to keep all three tables, they should collect the data in the same units of measurement so that the data are easily comparable between the baseline and post-intervention situation of the groups and between groups. 

It is recommended that these tables be redone as indicated. 

The information contained in L 426-428 does not seem to be consistent with the writing of the manuscript. 

 4. Discussion; 

It is structured and coherent. 

5. Conclusion: the conclusions are consistent with the evidence obtained and give an answer to the proposed objective.  

The statement "Although the control group showed similar results, the magnitude of reduction in the depression, anxiety, and stress scores of the intervention group was greater" is confusing in relation to the previous conclusion. It needs to be clarified or rewritten. 

Author Response

Response to Review date: 14 November 2022

The manuscript ijerph-2014252 "Health Education Module Based on Information-Motivation- Behavioural Skills (IMB) for Reducing Depression, Anxiety, and Stress Among Adolescents in Boarding Schools: A Clustered Randomised Controlled Trial" is a case-control study that analyzes the effectiveness of a mental health intervention in reducing the level of stress, anxiety, and depression among boarding school students.   

The manuscript is clear, brief, precise, relevant and novel. The objective is clearly and precisely stated and the results and discussion are developed in a structured way with the proposed objectives. In the results, it is necessary to adapt the information contained in the tables to facilitate comparison and reading. The discussion is presented in an orderly and well-integrated manner. 

It is a consistent and correctly developed manuscript, being, for the most part, easy to read. 

Nevertheless, a major revision of the present version before publication is recommended in order to include some suggestions for improvement, which mainly affect the clarification of the information contained in the tables of results. 

Specific remarks:

1- Introduction:

The wording is concise and in the right direction, from the concrete to the particular, all the necessary terms being defined. The research question is well stated. 

It would be convenient to include in the state of the question the data, if known, of adolescents with problems of depression, anxiety or stress during the pandemic, which complements the information provided on the prevalence of DAS included in the L47-58.

The statement on covid 19 pandemic has been added – L49 -50

It would also be enriching to indicate whether within the curriculum there is any subject in the schools in which the contents addressed in the intervention are addressed. 

Unfortunately, based on current limitation, the statement on the school curriculum is beyond author authority to judge or comment as this could give wrong or miss perception towards school. We are not able to provide the statement.

  1. Materials and methods: 

The material and methods are precisely described, as are the inclusion and exclusion criteria. 

2.3 Data collection process

Although this section of the manuscript reflects all the expected information, its reading is complex and extensive, making it difficult to follow the section. Although it is not usual in material and methods, it is recommended that a figure (diagram) be included to clarify the text or the rewrite so that it is easy to read.

Figure 1 is to explain the participant selection and data collection process.  

in this section it is also recommended to include how data collection is carried out on all participants at the same time (information included in later sections). 

Statement on data collection were added L252-255

2.4 Study instruments: 

L269-L272: "They were advised not to spend too much time on each statement. In addition, their knowledge on mental health and DAS, their attitude towards DAS, and self- 270 efficacy were assessed using another set of questionnaires adapted from different sources 271 (31-33)." 

The name of the questionnaires referenced in the text must be included (the bibliographic reference is not sufficient). 

We have decided to remove this statement as this statement actually refer the general advice by different paper on how to use DAS questionnaire and how to measure attitude and self-efficacy using different set of questionnaires (which not been used in this study)  

  1. Results:

The results in the text are adequately reflected.  Improvements in the presentation of the tables are recommended.  

Table 2 Initial comparison of depression, anxiety and stress scores; Table 3 Post-intervention comparisons of depression, anxiety and stress scores between groups; and Table 4 Post-intervention comparisons of depression, anxiety and stress scores within each group can be integrated into a single table. 

If it is desired to keep all three tables, they should collect the data in the same units of measurement so that the data are easily comparable between the baseline and post-intervention situation of the groups and between groups. 

It is recommended that these tables be redone as indicated. 

Each of table carry different analysis to answer specific objectives of this study. We are regret for not able to integrate the table as each table carry different analysis. For your information, it is easier to explain the finding base on the current table presentation and we hope you can accept our justification

The information contained in L 426-428 does not seem to be consistent with the writing of the manuscript. 

The information contained in L400-424 has been rewrite in consistent format.

  1. Discussion; 

It is structured and coherent. 

  1. Conclusion: the conclusions are consistent with the evidence obtained and give an answer to the proposed objective.  

The statement "Although the control group showed similar results, the magnitude of reduction in the depression, anxiety, and stress scores of the intervention group was greater" is confusing in relation to the previous conclusion. It needs to be clarified or rewritten.

This conclusion statement tells the truth of our study. From table 4 (comparison within each group), both group (intervention and control) show depression, anxiety and stress improvement. However, the effect of improvement is greater among intervention group. In other word, although control group (receive standard care) show improvement, intervention group (receive new intervention) show greater effect.  

Reviewer 2 Report

Thank you for the manuscript, please could you provide an outlook for further research at the end, and give an overview of the existing eHealth programms in this field, that are already existing, many thanks. 

Author Response

please could you provide an outlook for further research at the end,

-The recommendation for further action was added L516-545

give an overview of the existing eHealth programms in this field, that are already existing, many thanks

- No specific eHealth programme on DAS in specific group of population available. However, some eHealth promotion and education from various sources is available - mention in L148-154

Round 2

Reviewer 1 Report

The manuscript has improved enough for publication. The diagram presented in Material and Methods has clarified the section.  The results are now neat and with the right units to facilitate comparison and drawing conclusions.